# Spike S2 Subunit: The Dark Horse in the Race for Prophylactic and Therapeutic Interventions against SARS-CoV-2

**DOI:** 10.3390/vaccines9020178

**Published:** 2021-02-20

**Authors:** Kim Tien Ng, Nur Khairiah Mohd-Ismail, Yee-Joo Tan

**Affiliations:** 1Infectious Diseases Translational Research Program, Yong Loo Lin School of Medicine, National University of Singapore, Singapore 117545, Singapore; micnkt@nus.edu.sg (K.T.N.); micnkmi@nus.edu.sg (N.K.M.-I.); 2Department of Microbiology and Immunology, Yong Loo Lin School of Medicine, National University of Singapore, Singapore 117545, Singapore; 3Institute of Molecular and Cell Biology (IMCB), A*STAR, Singapore 138673, Singapore

**Keywords:** severe acute respiratory syndrome coronavirus 2, SARS-CoV-2, coronavirus disease 2019, COVID-19, spike glycoprotein, S2 subunit

## Abstract

In the midst of the unceasing COVID-19 pandemic, the identification of immunogenic epitopes in the SARS-CoV-2 spike (S) glycoprotein plays a vital role in the advancement and development of intervention strategies. S is expressed on the exterior of the SARS-CoV-2 virion and contains two subunits, namely the N-terminal S1 and C-terminal S2. It is the key element for mediating viral entry as well as a crucial antigenic determinant capable of stimulating protective immune response through elicitation of anti-SARS-CoV-2 antibodies and activation of CD4^+^ and CD8^+^ cells in COVID-19 patients. Given that S2 is highly conserved in comparison to the S1, here, we provide a review of the latest findings on the SARS-CoV-2 S2 subunit and further discuss its potential as an attractive and promising target for the development of prophylactic vaccines and therapeutic agents against COVID-19.

## 1. Introduction

In the past two decades, humankind has experienced the aftermath of two separate interspecies transmission events involving the highly pathogenic novel betacoronaviruses [1]. The first was severe acute respiratory syndrome coronavirus (SARS-CoV) that emerged in Foshan, Guangdong, China in November 2002, infecting more than 8000 individuals globally with a mortality rate of 10% [2]. Nearly 10 years after the containment of SARS-CoV, the Middle East respiratory syndrome coronavirus (MERS-CoV) was reported in June 2012 in Saudi Arabia [3], infecting over 2000 individuals and having an estimated case fatality rate of 35% [1]. The recent emergence of severe acute respiratory syndrome coronavirus 2 (SARS-CoV-2) that causes coronavirus disease 2019 (COVID-19) marked the third introduction of a highly pathogenic novel betacoronavirus of zoonotic origin into the human population [4]. SARS-CoV-2 was first observed in a cluster of patients with atypical viral pneumonia in Wuhan, Hubei Province, China in December 2019 [5]. Being highly transmissible with an estimated basic reproduction number (R_0_) of between 2.2 and 3.9 (as predicted from the initial outbreak in China) [6], the spread of SARS-CoV-2 through close contact and further mediated by international air-travel [4] has swiftly led to an unprecedented coronavirus pandemic with over 100 million cases and 2.16 million deaths globally, as of 29 January 2021. Clinical manifestations of COVID-19 vary broadly, ranging from asymptomatic infection [7] to respiratory failure and death [5].

Genomic characterization of SARS-CoV-2 revealed that this novel betacoronavirus is closely related to SARS-CoV but distantly related to MERS-CoV, sharing approximately 79% and 50% genome sequence identity, respectively [4]. With a genome structure common to other betacoronaviruses, SARS-CoV-2 has multiple open reading frames (ORFs) encoding for the accessory proteins, non-structural replicase (ORF1a/ORF1b), as well as the envelope (E), membrane (M), nucleocapsid (N), and spike (S) proteins. S, a structural glycoprotein expressed on the surface of SARS-CoV-2, is the key determinant of viral–host interaction and tissue tropism. This 180 kDa S glycoprotein is a crucial antigenic determinant capable of inducing protective immune responses. For instance, the elevation of anti-SARS-CoV-2 S neutralizing antibody titers has been reported in SARS-CoV-2-infected patients’ sera [8,9,10]. Likewise, the antigenicity of SARS-CoV-2 S is also apparent through the detection of S-specific CD4^+^ and CD8^+^ cells in blood specimens obtained from COVID-19 convalescents [9].

In response to the exigent need to develop safe and effective interventions against SARS-CoV-2, approximately 150 vaccine and antibody-based therapeutic approaches are currently being evaluated [11,12,13], with most of these approaches targeting the trimeric S protein. The S glycoprotein contains two subunits, namely the N-terminal S1 and C-terminal S2. The length of the SARS-CoV-2 S glycoprotein is 1273 amino acids (aa), arranged sequentially with a 13-aa signal peptide located at the N-terminus followed by the S1 subunit (residues 14–685) and the S2 subunit (residues 686–1273) [14]. Within the S1 subunit, there is an N-terminal domain (residues 14–305) and a receptor-binding domain (RBD; residues 319–541), whereas the fusion peptide (FP; residues 788–806), heptapeptide repeat sequence 1 and 2 (HR1; residues 912–984 and HR2; residues 1163–1213), transmembrane domain (TMD; residues 1213–1237), and cytoplasmic domain (residues 1237–1273) form the S2 subunit (Figure 1) [15]. Detailed characterization of the S1 subunit and its importance for binding to the angiotensin-converting enzyme 2 (ACE2) receptor have been described by various groups and summarized in recent reviews [14,16]. In comparison to the S1 subunit, the potential of the highly conserved SARS-CoV-2 S2 subunit that mediates viral–host membrane fusion to enable viral entry remains insufficiently explored. As such, this review focuses on our present understanding of S2 and discusses its relevance for the development of vaccines and therapeutic agents in combating the unceasing COVID-19 pandemic.

## 2. Structural and Functional Properties of the SARS-CoV-2 S2 Subunit

Like other coronaviruses, the S protein of SARS-CoV-2 facilitates host cell receptor recognition, cell attachment, and fusion during infection [17,18,19,20]. The S2 subunit, comprising FP, HR1, HR2, TMD, and cytoplasmic domains arranged sequentially, plays a crucial role in viral fusion and entry [14]. FP is a short 15–20 amino acid segment, consisting predominantly of hydrophobic glycine (G) or alanine (A) residues, which affix to the target membrane when the S protein assumes the prehairpin conformation. A previous study has demonstrated that FP plays an essential part in facilitating membrane fusion by disrupting and bridging the lipid bilayer of the host cell membrane [21]. On the other hand, HR1 and HR2 are constitutions of a repetitive HPPHCPC heptapeptide, where H is a hydrophobic residue, P is a polar hydrophilic residue, and C is another charged residue [22,23]. Together, HR1 and HR2 shape the 6-helical bundle (6HB) structure, which is imperative for effective viral fusion and cell entry function of the S2 subunit [15]. Structurally, HR1 is located at the C-terminus of the hydrophobic FP, and HR2 is located at the N-terminus of the TMD [24], with the downstream TMD anchoring the S protein to the viral membrane and the S2 subunit ending in a cytoplasmic tail [23].

Functionally, when RBD binds to ACE2, S2 changes its conformation by implanting FP into the target host cell membrane and revealing the prehairpin structure of the HR1 trimer to form a 6HB (Figure 2). This bridges the viral envelope and cell membrane into propinquity, ready for viral fusion and cell entry [25,26]. The HR1 forms a trimeric structure in which three highly conserved hydrophobic grooves on its surface that attach to HR2 are unmasked. Subsequently, the HR2 domain forms a rigid helix and a flexible loop to interact with the HR1 domain. Within this postfusion conformation, there are many strong interactions between the HR1 and HR2 domains inside the helical region, which is designated as the fusion core. A recent study has revealed the crystal structure of the SARS-CoV-2 fusion core and highlighted that the structure of the HR1 and HR2 complex of SARS-CoV-2 resembles those of SARS-CoV and MERS-CoV [27].

## 3. S2 Subunit as an Antiviral Target

Given that some regions within the S1 subunit are highly variable, S1 may not be the ideal target site for broad-spectrum anti-CoV inhibitor development [28]. On the contrary, the HRs embedded within the S2 subunit, which is highly conserved amongst the SARS-like CoV [29,30], has garnered enormous interest in therapeutic pan-coronaviral drug discovery [31].

Zhu and colleagues [32] used circular dichroism (CD) spectroscopy to study the secondary structure and thermal stability of HRs in SARS-CoV-2 S. The study revealed that, relative to SARS-CoV, the HR1 of SARS-CoV-2 exhibits significantly elevated α-helicity (by 22%) and thermostability (by 8 °C), coupled with a higher binding affinity to its corresponding HR2. Such heightened HR1–HR2 interaction potentially enhanced the efficiency of 6HB structure formation, resulting in an augmented fusion capacity. To demonstrate the potential of S2 as an important antiviral target, an HR2-specific sequence-based lipopeptide fusion inhibitor (IPB02) was designed, and it was shown to be highly potent in blocking S2-mediated cell fusion and pseudovirus transduction for both SARS-CoV and SARS-CoV-2.

Likewise, in a study by Xia and colleagues [33], the authors found that SARS-CoV-2 exhibited a more robust plasma membrane fusion capacity compared to SARS-CoV, implying the importance of targeting HR in the attempt to combat the virus. The authors went on to generate a series of lipopeptides based on the previously developed universal coronavirus fusion inhibitor (EK1), which was derived from the HR2 of the human coronavirus HCoV-OC43, and discovered that one of the derivatives, EK1C4, was the most potent fusion inhibitor against SARS-CoV-2 protein-mediated membrane fusion and pseudovirus infection, with IC_50_ values of 1.3 and 15.8 nM. Similarly, the potential of HR in S2 as a target for fusion inhibitors was also reported in other studies [27,34].

In a subsequent study, Outlaw and colleagues further compared EK1 with lipopeptides derived from the HR2s of SARS-CoV-2 and MERS-CoV to understand their potency and broad-spectrum activities [35]. Instead of a pseudotyped virus entry assay, the authors used a live virus-neutralizing assay to demonstrate that these three different lipopeptides can inhibit infection by both SARS-CoV-2 and MERS-CoV and found that the lipopeptide containing the HR2 of SARS-CoV-2 exhibited the most potent inhibition of SARS-CoV-2. Importantly, it was shown that this lipopeptide efficiently blocked SARS-CoV-2 spread in an ex vivo model using human airway epithelial cultures, which closely mimic the clinical scenario.

Based on these studies, which highlighted the importance of HR1–HR2 interaction, the S2 subunit is convincingly a desirable target for the development of small molecules or peptides as SARS-CoV-2 therapeutics. By preventing the formation of 6HB essential for membrane fusion, HR1- or HR2-derived peptidomimetics are effective in blocking viral entry in vitro as well as ex vivo and may act on different coronaviruses. Further in vivo assessments will be necessary to evaluate the efficacy of these peptidomimetics before advancing to clinical trials. Nevertheless, the advancement of peptidomimetics can be exploited to develop both SARS-CoV-2-specific therapy as well as those that are pan-CoV to enhance our pandemic preparedness [21].

## 4. S2-Specific Antibodies in Convalescent Sera from COVID-19 Patients Prevent Cell Entry of SARS-CoV-2 Spike-Pseudotyped and Live Virus Particles

The activation of humoral immune response and elicitation of epitope-targeted protective neutralizing antibodies serve as one of the primary defense mechanisms against viruses, where the antibodies inhibit infection by interfering with host–viral interactions and arresting the entry of viruses. Amid the ongoing COVID-19 pandemic, the identification of immunogenic targets within the SARS-CoV-2 S glycoprotein plays a pivotal role in the advancement and development of vaccine strategies. Overlapping peptides spanning the entire viral proteome have been used to identify linear B cell epitopes recognized by the human immune system. For instance, in a study by Yi and colleagues [36], the authors developed a peptide array comprising 20-mer overlapped peptides that span the entire S, the ectodomain of M, and E proteins. A total of 120 COVID-19 convalescent and 24 non-COVID-19 sera were screened, from which more than 40% of the COVID-19 sera reacted to 6 peptides that correspond to S (peptides S1–52, S1–55, S1–57, S2–13, and S2–47) and M (peptide M1). Importantly, by using a peptide-competition neutralization assay with live SARS-CoV-2, it was found that three of the S-associated dominant epitopes (S1–55, S1–57, and S2–47) partially blocked the neutralizing activity of COVID-19 sera, suggesting that antibodies elicited by these epitopes played an important role in neutralizing SARS-CoV-2.

In a study by Poh and colleagues [37], the antibody profiles of six COVID-19 convalescent patients were characterized, using pools of 18-mer overlapping linear peptides spanning the whole S glycoprotein. Two distinct immunodominant linear B-cell epitopes, namely S14P5 (residues 562–579, located close to the RBD in the S1 subunit) and S21P2 (residues 818–835 that flanks part of FP in the S2 subunit), on the S glycoprotein of SARS-CoV-2 were strongly detected by the sera. This observation was further validated with sera from 41 COVID-19 patients and 28 pre-pandemic healthy donors, from which the detection of both S14P5 and S21P2 were consistently and significantly higher in COVID-19 patients. Importantly, using an antibody depletion assay against S14P5 and S21P2 coupled with pseudotyped lentivirus expressing the SARS-CoV-2 S glycoprotein, the authors demonstrated that antibodies directed at these immunodominant sites (S14P5+S21P2) account for a significant fraction (~40%) of the total anti-S neutralizing response. When tested individually, the antibody depletion assay against the respective epitope yielded over 20% reduction each in pseudotyped lentivirus neutralization, substantiating the unorthodox importance of the S2 subunit in eliciting neutralizing antibodies. Interestingly, sera from two COVID-19 patients cross-reacted with a SARS-CoV peptide library that encompasses the highly conserved fusion peptide of the S2 subunit, warranting further assessment and highlighting the potential of S2 as a pan-CoV target.

Li and colleagues [38] also decoded the B-cell epitopes of SARS-CoV-2 S protein through a newly constructed 211-peptide microarray system and reported several epitopes in the S2 subunit that induce neutralizing antibodies in COVID-19 patients. A total of 55 sera from convalescent COVID-19 patients and 18 control sera were screened for both IgG and IgM responses. Compared to the IgM response, which is less informative and specific, the IgG response in COVID-19 patients was distinct from the control group and exhibited specific hotspot-defining signals. There were three prominent hotspots across the S protein, of which one was in the S1 subunit (S1: 93–113, residues 553–684 immediately downstream of RBD) and the other two were in the S2 subunit. The first S2 hotspot was S2: 14–23 (residues 764–829) that covers the fusion peptide and the S2 cleavage site (R815), with 11 residues overlapping the immunodominant epitope reported by Poh and colleagues [37]. It was noted that IgG responses against the different epitopes within the S2: 14–23 hotspot poorly correlated with each other, potentially driven by the tendency of this region to form continuous but competitive epitopes. Additionally, this region is partially shielded by other structures in prefusion trimeric S, highlighting that it may only be physically accessible to the immune system in the fusogenic form of S. The second hotspot in the S2 subunit was S2–78 (residues 1148–1159) that links heptad repeat 1 (HR1) and heptad repeat 2 (HR2). Remarkably, IgG targeting this epitope was detected in 90% of COVID-19 patients, accentuating the immunodominance of this epitope. Furthermore, antibodies against this S2–78 epitope, separately enriched from five sera, displayed high specificity and potent neutralizing activity, with pseudoviral inhibitory efficiency of 35% at 21µg/ml. Sequence analysis of the S2-associated immunogenic epitopes (S21P2 and S2–78) that induce neutralizing antibodies in patients revealed that these epitopes are indeed highly conserved between SARS-CoV, MERS-CoV, and HCoV-OC43 and -HKU1, with higher divergence observed in human alphacoronaviruses (HCoV-229E and -NL63) (Table 1). Thus, it is apparent that the S2 subunit may be a better target for the development of broad-spectrum prophylactic or therapeutic agents.

These peptide-based studies demonstrated that anti-S2 antibodies are being stimulated in COVID-19 patients and could contribute to the total level of neutralizing serum activities. One limitation of using peptides is that conformationally dependent anti-S2 antibodies will not be identified in these studies. However, the S1 subunit containing the RBD is likely to stimulate the majority of neutralizing antibodies in COVID-19 patients and almost all the highly potent neutralizing monoclonal antibodies isolated from COVID-19 convalescent patients were found to bind to the S1 subunit [39,40]. This may primarily be due to the structural feature of S, which is extensively protected from antibody recognition by glycans, with the prominent exception of the RBD, thus contributing to the observed immunodominance of RBD epitopes [41]. However, with the recent emergence of SARS-CoV-2 variants that harbor multiple mutations at the highly variable RBD and capable of escaping host immune responses, the highly conserved S2 subunit (Table 1) may be a better target for broad-spectrum potent neutralizing monoclonal antibodies. Thus far, there are two anti-S2 neutralizing monoclonal antibodies that have been isolated from COVID-19 patients and reported in preprints, of which one demonstrated the ability of an S2 monoclonal antibody in neutralizing the emerging variant [42,43].

## 5. SARS-CoV-2 S2-Reactive T and Memory B Cells

In addition to humoral immunity, cellular immunity serves as another line of defense against viral infection. Compared to humoral immunity, cellular immunity, as observed in patients previously infected with SARS-CoV, is long-lived with SARS-CoV-specific memory T-cells persisting in the peripheral blood mononuclear cells (PBMCs) of SARS-CoV convalescents for up to 17 years post infection [44]. In general, antigen-specific CD4^+^ and CD8^+^ T cells exist at exceptionally low levels in naïve hosts. Upon active infection or immunization, the naïve T cells undertake clonal proliferation, culminating in a higher number of antigen-specific cells with swift effector functions. These alterations in T cell numbers and function constitute immune memory, and these memory T cells, in concert with antibody responses, form the foundation for protective immunity against COVID-19 [45,46].

In a study by Grifoni and colleagues [47], the authors utilized HLA classes I and II predicted peptide megapools to analyze SARS-CoV-2-specific T cell responses in 10 patients with COVID-19 and 11 healthy SARS-CoV-2-unexposed control participants. While circulating SARS-CoV-2 S-targeted and non-S-targeted CD4^+^ cells were identified in 100% of COVID-19 convalescents, SARS-CoV-2-specific CD8^+^ T cell responses were only observed in 70% of COVID-19 convalescent patients. However, a significant proportion of this CD8^+^ T cell reactivity (~26%) was derived from spike epitopes, highlighting the antigenicity of the S protein in inducing T-cell responses and its potential for prophylactic vaccination strategies [48,49,50]. Interestingly, Grifoni and colleagues also detected SARS-CoV-2-reactive CD4^+^ and CD8^+^ T cells in 60% and 40% of SARS-CoV-2-unexposed healthy donors, respectively. These observations suggest that the recognition of SARS-CoV-2 antigens by pre-existing cross-reactive T cells, induced by previous endemic human coronaviruses (hCoVs-OC43, -HKU1, -NL63, and -229E) infections, may not be uncommon.

Likewise, in a study by Braun and colleagues [51] that involved 68 SARS-CoV-2-unexposed healthy donors and 25 PCR-confirmed COVID-19 patients, the authors also identified S glycoprotein-reactive CD4^+^ T cells in 35% of the healthy participants. Among the COVID-19 patients, SARS-CoV-2 S-reactive CD4^+^ T cells were identified in 15 out of 18 patients (83%). This study utilized two pools of 15-aa overlapping S peptides for in vitro simulation, with one pool consisting of S-I peptides derived from the N-terminal region and the other pool containing S-II peptides from the C-terminal of S. Unlike the primarily S2-reactive CD4^+^ T cell response observed in healthy donors, the CD4^+^ T cells from COVID-19 patients equally targeted the S-I and S-II peptide pools. Notably, it was observed that S2-reactive CD4^+^ T cells from both COVID-19 patients and healthy donors exhibited a memory phenotype and significant TH1 polarization, characterized by the expression of IFNɣ.

Based on the aforementioned studies that examined T cell immune responses to SARS-CoV-2, it is therefore not surprising if SARS-CoV-2-reactive memory B cells (MBCs) are present in SARS-CoV-2-unexposed individuals due to their previous infections by endemic human coronaviruses. In a study by Nguyen-Contant and colleagues [52], the authors compared antibodies and MBC immunities to SARS-CoV-2 by analyzing sera and PBMCs from 21 SARS-CoV-2-unexposed participants and 26 COVID-19 convalescent patients. They found that 86% of SARS-CoV-2-unexposed subjects possess IgG that predominantly targets the highly conserved S2 subunit of the S protein, although a significantly higher IgG level against S2 was observed in convalescent patients. Notably, serum IgG titers targeting S2 were consistently higher than IgG against the RBD in convalescent patients, suggesting reactivation of previously generated anti-S2 IgG MBCs as opposed to activation of naïve B cells against the novel SARS-CoV-2 RBD. Separately, PBMCs from the study participants were analyzed for the presence of MBCs reactive to SARS-CoV-2 S proteins by in vitro stimulation of MBCs to induce differentiation into antibody-secreting cells (ASCs). The study revealed that a vast majority of the convalescent subjects had circulating IgG MBCs reactive to the SARS-CoV-2 spike. Notably, S2-reactive IgG MBC numbers correlated well with levels of IgG MBCs reactive to the SARS-CoV-2 spike (*P* < 0.0001). Since MBCs are long-lived cells, the study highlights the possibility of sustained protection by IgG MBCs even when circulating antibody levels wane over time. Importantly, IgG MBCs are readily activated to generate strong antibody responses or seed germinal centers for additional rounds of affinity maturation.

## 6. Cross-Reactive Immunity against Coronavirus S2 Subunit

SARS-CoV-2 infection yields different clinical manifestations according to age, with symptomatic and severe SARS-CoV-2-associated complications predominantly observed in older adults [53]. Such disparity could potentially be due to children and younger adults having a higher level of pre-existing immunity against endemic human coronaviruses (HCoVs), namely HCoV-OC43, -HKU1, -NL63, and -229E, which are able to cross-react with SARS-CoV-2 [54]. In line with this, SARS-CoV-2 cross-reactive memory CD4^+^ and CD8^+^ T cells against the S protein have been reported elsewhere in a significant proportion of SARS-CoV-2-unexposed subjects [47,51,55]. Notably, Braun and colleagues [51] reported that the S-specific CD4^+^ T cell response in healthy donors was mainly directed against the S-II peptide pool, which is consistent with the higher homology observed in the C-terminal region (S2) of the SARS-CoV-2 and endemic human coronaviruses S glycoprotein. While the role of these pre-existing cross-reactive T cells in SARS-CoV-2-naïve individuals remains elusive, it is conjectured that the presence or absence of these cells might contribute to the different clinical manifestations and outcome of COVID-19.

Interestingly, emerging bodies of evidence suggest that humoral immunity against SARS-CoV-2 in COVID-19 patient sera may also cross-react with the endemic HCoVs. For instance, in an attempt to profile the SARS-CoV-2 epitopes in COVID-19 patients using VirScan, Shrock and colleagues [56] reported strong recognition of endemic HCoV peptides by COVID-19 patient sera, possibly through the activation of existing MBCs against these HCoVs or generation of new cross-reactive antibodies. Consistent with other studies, the SARS-CoV-2-unexposed sera profiled here showed cross-reactive antibody responses against an array of coronaviruses, including SARS-CoV-2. Importantly, it was noted that pre-existing antibodies in the SARS-CoV-2-unexposed subjects recognized non-structural proteins, whereas only COVID-19 patients’ antibodies primarily recognized S and N structural proteins of SARS-CoV-2.

In addition, using a sensitive flow cytometry-based assay, Ng and colleagues [57] revealed that SARS-CoV-2-unexposed subjects, especially children and adolescents, possessed neutralizing antibodies predominantly of the IgG class and targeted the S2 subunit. On the contrary, SARS-CoV-2 infection elicited higher titers of SARS-CoV-2 S-reactive IgG antibodies against both the S1 and S2 subunits and concomitant IgM and IgA antibodies. The class-switching of cross-reactive antibodies to the mature IgG and IgA observed in SARS-CoV-2-unexposed subjects suggested that B cells producing these cross-reactive antibodies were induced by previous immune response against endemic common cold HCoV infections. The above studies substantiated the existence of cross-reactive humoral immunity against the conserved S2 subunit of HCoVs and SARS-CoV-2, thereby highlighting the possibility of pan-CoV intervention targeting the S2 subunit. On the other hand, the presence of cross-reactive antibodies also raises concern of possible antibody-dependent enhancement (ADE) of SARS-CoV-2 infection by non-neutralizing or subeffective levels of antibodies [58,59]. Due to insufficient current clinical data, the contribution of ADE to COVID-19 disease pathology and the role of pre-existing cross-reactive antibodies in clinical ADE of SARS-CoV-2 are still largely unknown.

## 7. Challenges and Potential Strategies for Targeting S2

Generation of antibodies with broad neutralizing activity or the development of a pan-coronavirus fusion inhibitor against the S2 subunit of different coronaviruses, or at least SARS-like coronaviruses, would be of great value for confronting future waves of coronavirus-associated diseases, especially when the risk of coronaviral spillover is inevitable [60]. However, the S2 conformation is highly dynamic during membrane fusion, presenting a major challenge in discovering antibodies against this protein subunit [61]. In this aspect, the potential of antigen-stabilizing strategies deployed during the discovery of antibodies against HIV and respiratory syncytial virus proteins may be explored to design stable coronavirus S2 proteins [62,63]. The S2 conformation selected as the target for vaccine development is also crucial as the postfusion conformation may expose non-neutralizing epitopes which distract the host immunity, as previously highlighted in a review article [64].

Synthetic peptides that can induce immune responses to specific antigenic S2 epitopes form another attractive avenue. One of the advantages of peptide vaccines is that they can induce broad immunity against multiple viral variants if the peptides are generated based on highly conserved epitopes. Importantly, peptide vaccines can also be synthesized to encompass multiple conserved epitopes, either observed in COVID-19 convalescents or through in silico prediction, to further enhance its capability as a pan-coronaviral vaccine that targets multiple serological variants or strains [65]. For instance, Flu-V is a multiple-epitope peptide vaccine that comprises four peptides representing CD8^+^ T-cell epitopes derived from the conserved regions in internal proteins M1, NPA, NPB, and M2 of Influenza A and Influenza B. A single dose of Flu-V with adjuvant led to a significant reduction in the number of influenza symptoms experienced by 40% of participants compared to those receiving placebo [66].

The conjugation of multiple SARS-CoV-2-derived immunogens to ferritin nanoparticles may be another potential vaccine alternative [67]. It has been reported that ferritin nanoparticles conjugated with 70% of RBD and 30% of HR subunits (known as RBD-HR hereafter) were able to elicit abundant S-specific neutralizing antibodies, conferring hACE2-mice’s total protection from SARS-CoV-2 infection upon authentic virus challenge. Interestingly, only neutralizing antibodies induced by RBD-HR nanoparticles, as opposed to those induced by RBD nanoparticles, were able to neutralize other coronaviruses, underlining the importance of S2 in the search for pan-CoV prophylactics and therapeutics.

## 8. Conclusions

Antigenic drift has been observed among endemic HCoV-OC43 [68,69], HCoV-229E [70], and in SARS-CoV [71,72]. Although there is no strong evidence that substantiates antigenic shift for SARS-CoV-2 [73], the persistence of the COVID-19 pandemic and extended human-to-human transmission may enable SARS-CoV-2 to acquire mutations conferring fitness advantages. Th emergence of SARS-CoV-2 B.1.1.7 (also known as 501Y.V1), as reported by the COVID-19 Genomics UK Consortium on 19 December 2020, generated a great shock wave globally during its initial discovery as this newly emerging variant that harbors 17 mutations was associated with substantially increased transmissibility while the impact of these mutations on disease prognosis and vaccine efficacy was largely unknown [74]. Of the 17 mutations, 8 (47%) mutations were observed in S, with N501Y substitution at the RBD and deletion at position 69–70 (∆69–70) of the furin cleavage site garnering the most attention, as preprints have highlighted that these mutations increase binding affinity to the ACE2 receptor and promote evasion of the host immune response, respectively [75,76]. Two other RBD mutations (N439K and Y453F) also conjured grave concern as preprints have indicated that these mutations also increase binding affinity to ACE2 and enable the variant to escape neutralization by a few monoclonal antibodies [77,78]. In comparison to the S1 subunit, the S2 subunit of coronavirus is more conserved [29,57], with higher chance of bearing epitopes targeted by broadly neutralizing antibodies [17,56,57]. In a nutshell, the importance of the highly conserved S2 subunit as a promising target for prophylactic and therapeutic strategies is increasingly evident. However, the precise selection of the immunodominant site remains crucial to fully uncover the benefits of targeting the S2 subunit.

## Figures and Tables

**Figure 1 vaccines-09-00178-f001:**
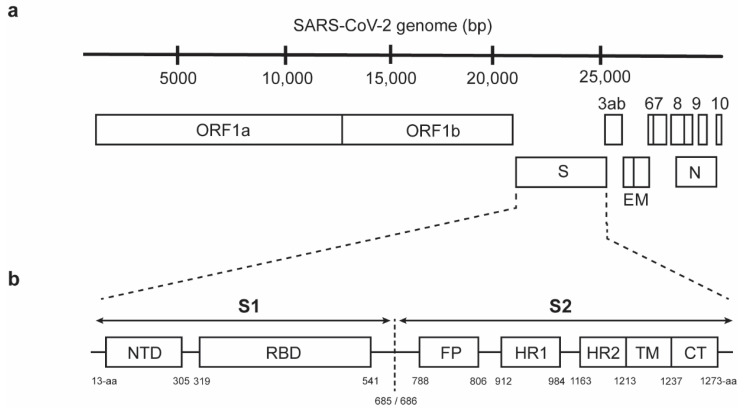
Schematic representation of (**a**) the genome structure (in bp) of severe acute respiratory syndrome coronavirus 2 (SARS-CoV-2) that is causing the coronavirus disease 2019 (COVID-19) pandemic and (**b**) the spike (S) protein (in aa) of SARS-CoV-2, comprising the S1 and S2 subunits. The residue numbers of each region correspond to the position in the S protein. ORF, open reading frame; S, spike; E, envelop; M, membrane; N, nucleocapsid, NTD, N-terminal domain; RBD, receptor binding domain; FP, fusion peptide; HR1, heptad repeat 1; HR2, heptad repeat 2; TM, transmembrane domain; CT, cytoplasmic tail.

**Figure 2 vaccines-09-00178-f002:**
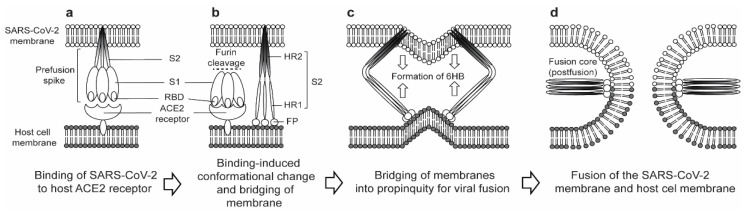
Schematic representation of (**a**) binding of SARS-CoV-2 prefusion spike (S) RBD to host ACE2 receptor, (**b**) cleavage of the S protein into the S1 subunit and the S2 subunit by furin, resulting in the fusion peptide (FP) of S2 being exposed and implanted into the target cell membrane, (**c**) formation of the 6-helical bundle (6HB) that bridges the membranes into propinquity for viral fusion, and (**d**) three HR1s and HR2s combining to form the fusion core (postfusion) to fuse the viral–host membrane.

**Table 1 vaccines-09-00178-t001:** The S and its component sequence identity between SARS-CoV-2 and other human coronaviruses.

	SARS-CoV-2 (YP009724390)					
	Amino Acid Sequence Identity					
	Full Length S	S1	RBD	S2	S21P2 ^a^	S2–78 ^b^
	1273 aa	685 aa	319–541 aa	686–1273 aa	818–835 aa	1148–1159 aa
SARS-CoV (YP009825051)	76.9%	65.2%	74.7%	89.3%	88.9%	99.4%
MERS-CoV (NC019843)	31.0%	20.5%	18.1%	41.9%	61.1%	63.0%
HCoV-OC43 (YP009555241)	28.6%	19.7%	20.3%	37.5%	72.2%	75.0%
HCoV-HKU1 (YP173238)	27.0%	18.8%	17.6%	35.3%	61.1%	50.0%
HCoV-229E (NP073551)	26.5%	15.3%	15.9%	34.6%	50.0%	16.7%
HCoV-NL63 (YP003767)	24.9%	15.9%	18.5%	33.3%	55.6%	8.3%

S21P2 ^a^ and S2–78 ^b^ are highly immunogenic in convalescent patients and capable of stimulating neutralizing antibodies [37,38].

## Data Availability

Not applicable.

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
