# Peer review of "Spike S2 Subunit: The Dark Horse in the Race for Prophylactic and Therapeutic Interventions against SARS-CoV-2"

_vaccines, 2021, doi:10.3390/vaccines9020178_

Round 1

Reviewer 1 Report

Vaccines manuscript review 1110764                                                                                                             02.09.21

Summary: The manuscript seeks to review knowledge and demonstrate the potential of the S2 subunit of the SARS-CoV-2 spike protein for its immunogenicity, and target for vaccines and therapeutics. This is a timely and important topic, and the manuscript serves as a resource to highlight the potential importance of the S2 subunit in vaccine/therapeutic strategies against SARS-CoV-2 and to summarize what is known so far regarding the role of S2.

Comments: The manuscript is clear and organized, presenting relevant literature. There are some areas where some greater depth of analysis/discussion would enhance the manuscript and its usefulness to the reader (details below). I would like to see you provide a stronger rationale to the reader of why evaluation of S2 is important -  given we already know S1 can provide a protective immune response – it would be great to see you provide stronger rationale for why S2 may be critical for broad/cross reactive response or as target. It would be beneficial to mention that we don’t yet fully appreciate correlates of immunity for this virus. You hinted at the utility of inducing pan-CoV immunity or targets for therapy and set up nicely that we’ve had 3 major introductions of lethal viruses in humans from animals but didn’t strongly conclude how this preparedness would be a tool against new/future emergence of related coronaviruses that may/will happen in the future – this might be a nice aspect to include.

Detailed edits (with line reference):

38-39: Please better describe the source of R0 range cited in terms of which period of time (initial outbreak versus current spread). R0 differs with time and geographic region, so giving this range is somewhat lacking without some context and purpose for it.

51-56: The statement is made that “S glycoprotein is a crucial antigenic determinant capable of inducing protective immune responses” however the two sentences that follow make the point that S is antigenic but doesn’t provide any evidence for the statement that S induces a protective immune response. Role of S in protection should be addressed.

59: Use of the word ‘scrutinized’ here is not incorrect but seems unnecessarily emotive- a more descriptive/scientific term would be more appropriate such as ‘evaluated’ or ‘tested’

70-73: I think you could revise this to present the rationale/logic and purpose of this review more clearly.

101: Figure 1 is great to clearly show the components (subunits and motifs) of S protein. If possible, it would be interesting and helpful to see a diagram of S protein structure or conformation and steps/forms it utilizes for cellular entry (to highlight the point about trimers and accessibility to regions in prefusion state).

135: This is purely for style/preference, but I would prefer use of ‘colleagues’ then ‘coworkers’.

147-148: Would like to see some discussion here regarding need or next steps to test in vivo to further evaluate potential or to advance to clinical trials.

151-152: The use of ‘COVID-19 convalescents’ is clunky, consider ‘convalescent sera from COVID-19 patients’

199-201: This could be explained better – what poorly correlated with what?

201-202: Could clarify that this means is physically inaccessible in prefusion. Consider use of ‘structures’ rather than ‘parts’ and ‘shielded’ rather than ‘safegarded’

204-206: In relation to immunodominance of the HR1-HR2 linking epitope – it would be worth providing some context for how variable/conserved is this region in SARS-CoV-2 and across other coronaviruses?

212-217: This gives an opportunity to provide some thoughtful discussion/analysis on whether conformation/structure is more critical in the RBD, or whether it is likely to be under greater immune selection and why S2 may represent a good target due to conservation and potential for cross-reactivity against variants.

302: edit typo ‘SAR-CoV-2’ to ‘SARS-CoV-2’

322: Please expand/describe for the reader what you mean by ‘generated a great shock wave globally’, or in what way did this occur.

330-332: discuss that S2 is more conserved – I think this is a critical point in linking the possible utility of S2 in potential for cross-protection, pan-CoV treatment, pandemic preparedness and would really like to see this emphasized at the beginning of the manuscript as a reason for why it may represent an important target (and hence why the review is of importance)

344-348: the concept of serotypes is mentioned generically – but as different serotypes are not recognized for this virus, it is a little confusing. Try to clarify the generic idea versus what is known for SARS-CoV-2 variants.

364: With the discussion and themes through the manuscript of antigens/ antibodies/ neutralization/ cross-reactivity, it is not appropriate to introduce the concept of ADE in the last couple of lines of the conclusion – this needs to be described earlier, particularly in the context of cross-reactivity to other human coronaviruses that people have likely been exposed to. Perhaps touch on what we know (or don’t yet know) about levels of SARS-CoV-2 immune measures required for protection, and implications for sub-effective levels or specificity. The reference #71 cited showed no evidence for ADE.

Author Response

Dear colleague,

Thank you so much for sharing your constructive comments! We have done the necessary revision and believe your comments have significantly improved the quality of the article. Thank you, again!

REVIEWER 1

Summary: The manuscript seeks to review knowledge and demonstrate the potential of the S2 subunit of the SARS-CoV-2 spike protein for its immunogenicity, and target for vaccines and therapeutics. This is a timely and important topic, and the manuscript serves as a resource to highlight the potential importance of the S2 subunit in vaccine/therapeutic strategies against SARS-CoV-2 and to summarize what is known so far regarding the role of S2.

Comments: The manuscript is clear and organized, presenting relevant literature. There are some areas where some greater depth of analysis/discussion would enhance the manuscript and its usefulness to the reader (details below). I would like to see you provide a stronger rationale to the reader of why evaluation of S2 is important - given we already know S1 can provide a protective immune response – it would be great to see you provide stronger rationale for why S2 may be critical for broad/cross reactive response or as target. It would be beneficial to mention that we don’t yet fully appreciate correlates of immunity for this virus. You hinted at the utility of inducing pan-CoV immunity or targets for therapy and set up nicely that we’ve had 3 major introductions of lethal viruses in humans from animals but didn’t strongly conclude how this preparedness would be a tool against new/future emergence of related coronaviruses that may/will happen in the future – this might be a nice aspect to include.

Detailed edits (with line reference):

Comment 1: 38-39: Please better describe the source of R0 range cited in terms of which period of time (initial outbreak versus current spread). R0 differs with time and geographic region, so giving this range is somewhat lacking without some context and purpose for it.

Response 1: We have added “as predicted from the initial outbreak in China” (Line 40)

Comment 2: 51-56: The statement is made that “S glycoprotein is a crucial antigenic determinant capable of inducing protective immune responses” however the two sentences that follow make the point that S is antigenic but doesn’t provide any evidence for the statement that S induces a protective immune response. Role of S in protection should be addressed.

Response 2: We have added the word “neutralizing” (Line 55) and added new references that provide evidence that S induces a protective immune response.

Comment 3: 59: Use of the word ‘scrutinized’ here is not incorrect but seems unnecessarily emotive- a more descriptive/scientific term would be more appropriate such as ‘evaluated’ or ‘tested’

Response 3: We have replaced “scrutinized” with “evaluated” (Line 61).

Comment 4: 70-73: I think you could revise this to present the rationale/logic and purpose of this review more clearly.

Response 4: We have amended the point with “In comparison to the S1 subunit, potential of the highly conserved SARS-CoV-2 S2 subunit that mediates viral-host membrane fusion to enable entry remains insufficiently explored. As such, this review focuses on our present understanding of S2 and discusses its relevance for the development of vaccines and therapeutic agents in combating the unceasing COVID-19 pandemic” (Line 72-77).

Comment 5: 101: Figure 1 is great to clearly show the components (subunits and motifs) of S protein. If possible, it would be interesting and helpful to see a diagram of S protein structure or conformation and steps/forms it utilizes for cellular entry (to highlight the point about trimers and accessibility to regions in prefusion state).

Response 5: We have added new Figure 2 (Page 3), as suggested.

Comment 6: 135: This is purely for style/preference, but I would prefer use of ‘colleagues’ then ‘coworkers’.

Response 6: We have replaced “coworkers” with “colleagues” (Line 150).

Comment 7: 147-148: Would like to see some discussion here regarding need or next steps to test in vivo to further evaluate potential or to advance to clinical trials.

Response 7: We have added “Further in vivo assessments will be necessary to evaluate the efficacy of these peptidomimetics before advancing to clinical trials” (Line 163-165)

Comment 8: 151-152: The use of ‘COVID-19 convalescents’ is clunky, consider ‘convalescent sera from COVID-19 patients’

Response 8: We have edited to “S2-specific Antibodies in Convalescent Sera from COVID-19 Patients Prevent Cell Entry of SARS-CoV-2 Spike-pseudotyped and Live Virus Particles” (Line 168-169)

Comment 9: 199-201: This could be explained better – what poorly correlated with what?

Response 9: We have restructured the sentence to “It was noted that IgG responses against the different epitopes within the S2: 14–23 hotspot poorly correlated with each other, potentially driven by the tendency of this region to form continuous but competitive epitopes.” (Line 216-218).

Comment 10: 201-202: Could clarify that this means is physically inaccessible in prefusion. Consider use of ‘structures’ rather than ‘parts’ and ‘shielded’ rather than ‘safegarded’

Response 10: Done with necessary amendment and updated the sentence to “Additionally, this region is partially shielded by other structures in prefusion trimeric S, highlighting that it may only be physically accessible to the immune system in the fusogenic form of S.” (Line 218-221).

Comment 11: 204-206: In relation to immunodominance of the HR1-HR2 linking epitope – it would be worth providing some context for how variable/conserved is this region in SARS-CoV-2 and across other coronaviruses?

Response 11: We have added Table 1 and added “Sequence analysis of the S2-associated immunogenic epitopes (S21P2 and S2-78) that induce neutralizing antibodies in patients revealed that these epitopes are indeed highly conserved between SARS-CoV, MERS-CoV, HCoV-OC43 and -HKU1, with higher divergence observed in human alphacoronaviruses (HCoV-229E and -NL63) (Table 1). Thus, it is apparent that the S2 subunit may be a better target for the development of broad-spectrum prophylactic or therapeutic agents.” (Line 226-232).

Comment 12: 212-217: This gives an opportunity to provide some thoughtful discussion/analysis on whether conformation/structure is more critical in the RBD, or whether it is likely to be under greater immune selection and why S2 may represent a good target due to conservation and potential for cross-reactivity against variants.

Response 12: We have added “This may primarily be due to the structural feature of S, which is extensively protected from antibody recognition by glycans, with the prominent exception of the RBD thus contributing to the observed immunodominance of RBD epitopes. However, with the recent emergence of SARS-CoV-2 variants that harbor multiple mutations at the highly variable RBD and capable of escaping host immune responses, the highly conserved S2 subunit (Table 1) may be a better target for broad-spectrum potent neutralizing monoclonal antibodies” (Line 245-252).

Comment 13: 302: edit typo ‘SAR-CoV-2’ to ‘SARS-CoV-2’

Response 13: Thank you for pointing out. We have corrected the error (Line 338)

Comment 14: 322: Please expand/describe for the reader what you mean by ‘generated a great shock wave globally’, or in what way did this occur.

Response 14: We edited to “Emergence of SARS-CoV-2 B.1.1.7 (also known as 501Y.V1), as reported by the COVID-19 Genomics UK Consortium on 19th December 2020, generated a great shock wave globally during its initial discovery as this newly emerging variant which harbors 17 mutations was associated with substantially increased transmissibility while the impact of these mutations on disease prognosis and vaccine efficacy were largely unknown” (Line 399-404).

Comment 15: 330-332: discuss that S2 is more conserved – I think this is a critical point in linking the possible utility of S2 in potential for cross-protection, pan-CoV treatment, pandemic preparedness and would really like to see this emphasized at the beginning of the manuscript as a reason for why it may represent an important target (and hence why the review is of importance)

Response 15: We could not agree more. We have added “Given that S2 is highly conserved in comparison to the S1, here we provide a review of the latest findings on the SARS-CoV-2 S2 subunit and further discuss its potential as an attractive and promising target for the development of prophylactic vaccines and therapeutic agents against COVID-19” (Abstract line 19-22), and “In comparison to the S1 subunit, potential of the highly conserved SARS-CoV-2 S2 subunit that mediates viral-host membrane fusion to enable viral entry remains insufficiently explored. As such, this review focuses on our present understanding of S2 and discusses its relevance for the development of vaccines and therapeutic agents in combating the unceasing COVID-19 pandemic.” (Introduction line 72-77)

Comment 16: 344-348: the concept of serotypes is mentioned generically – but as different serotypes are not recognized for this virus, it is a little confusing. Try to clarify the generic idea versus what is known for SARS-CoV-2 variants.

Response 16: We have replaced “serotypes” with “variants” (Line 372)

Comment 17: 364: With the discussion and themes through the manuscript of antigens/ antibodies/ neutralization/ cross-reactivity, it is not appropriate to introduce the concept of ADE in the last couple of lines of the conclusion – this needs to be described earlier, particularly in the context of cross-reactivity to other human coronaviruses that people have likely been exposed to. Perhaps touch on what we know (or don’t yet know) about levels of SARS-CoV-2 immune measures required for protection, and implications for sub-effective levels or specificity. The reference #71 cited showed no evidence for ADE.

Response 17: Thank you for suggestion. We have moved and rewritten the paragraph for better clarity. As discussed in several recent reviews, the contribution of ADE to COVID-19 disease pathology and the role of pre-existing cross-reactive antibodies in clinical ADE of SARS-CoV-2 are still largely unknown. Thus, we have added these citations and removed reference #71. (Line 350-355)

Reviewer 2 Report

This is an excellent review which provides a new perspective on a potential target spike 2 subunit(S2) for SARS vaccine and therapeutic development.  It is comprehensive and thoroughly listed the structure and function of S2, the evidence that S2 can stimulate both humoral immunity and cellular immunity, as well as the cross-reactive immunity among human coronaviruses. All above suggested S2 is a potential antiviral target. Some comments may help to improve this manuscript as below:

  1. The authors pointed out the challenge and limit in develop antibodies against S2 in conclusion. In my opinion, this part is worth to be stated and emphasized individually. There are remaining limits of S2 as the target to design vaccines (g. the postfusion conformation may expose the non-neutralizing epitopes and distract the host immunity. Yetian Dong, 2020). Those also need to be presented.
  2. In addition, the authors listed some strategies: peptide vaccines and ferritin nanoparticles conjugation.Those also worth to be raised in a separate ‘potential strategies’ part, not in conclusion.
  3. There are many vaccines based on spike protein of SARS-CoV in development, especially based on RBD and full-length S protein. Could it be advisable to compare S2 with those targets and point out why S2 target stand out among those targets?

Author Response

Dear colleague,

Thank you so much for sharing your constructive comments! We have done the necessary revision and believe your comments have significantly improved the quality of the article. Thank you, again!

REVIEWER 2

This is an excellent review which provides a new perspective on a potential target spike 2 subunit(S2) for SARS vaccine and therapeutic development.  It is comprehensive and thoroughly listed the structure and function of S2, the evidence that S2 can stimulate both humoral immunity and cellular immunity, as well as the cross-reactive immunity among human coronaviruses. All above suggested S2 is a potential antiviral target. Some comments may help to improve this manuscript as below:

Comment 1: The authors pointed out the challenge and limit in develop antibodies against S2 in conclusion. In my opinion, this part is worth to be stated and emphasized individually. There are remaining limits of S2 as the target to design vaccines (g. the postfusion conformation may expose the non-neutralizing epitopes and distract the host immunity. Yetian Dong, 2020). Those also need to be presented.

Response 1: Thank you so your suggestion. We have created Section 7 “Challenges and Potential Strategies for Targeting S2” (Line 357). We have also added “The S2 conformation selected as the target for vaccine development is also crucial as the post-fusion conformation may expose non-neutralizing epitopes which distract the host immunity, as previously highlighted in a review article” (Line 366-369)

Comment 2: In addition, the authors listed some strategies: peptide vaccines and ferritin nanoparticles conjugation. Those also worth to be raised in a separate ‘potential strategies’ part, not in conclusion.

Response 2: We have created Section 7 – “Challenges and Potential Strategies for Targeting S2” (Line 357).

Comment 3: There are many vaccines based on spike protein of SARS-CoV in development, especially based on RBD and full-length S protein. Could it be advisable to compare S2 with those targets and point out why S2 target stand out among those targets?

Response 3: Thank you for your suggestion. We have added Table 1 that compares these targets.